# Efficient Long-Range Convolutions for Point Clouds

## Abstract

The efficient treatment of long-range interactions for point clouds is a challenging problem in many scientific machine learning applications. To extract global information, one usually needs a large window size, a large number of layers, and/or a large number of channels. This can often significantly increase the computational cost. In this work, we present a novel neural network layer that directly incorporates long-range information for a point cloud. This layer, dubbed the long-range convolutional (LRC)-layer, leverages the convolutional theorem coupled with the non-uniform Fourier transform. In a nutshell, the LRC-layer mollifies the point cloud to an adequately sized regular grid, computes its Fourier transform, multiplies the result by a set of trainable Fourier multipliers, computes the inverse Fourier transform, and finally interpolates the result back to the point cloud. The resulting global all-to-all convolution operation can be performed in nearly-linear time asymptotically with respect to the number of input points. The LRC-layer is a particularly powerful tool when combined with local convolution as together they offer efficient and seamless treatment of both short and long range interactions. We showcase this framework by introducing a neural network architecture that combines LRC-layers with short-range convolutional layers to accurately learn the energy and force associated with a $N$-body potential. We also exploit the induced two-level decomposition and propose an efficient strategy to train the combined architecture with a reduced number of samples.

## 1 Introduction

Point-cloud representations provide detailed information of objects and environments. The development of novel acquisition techniques, such as laser scanning, digital photogrammetry, light detection and ranging (LIDAR), 3D scanners, structure-from-motion (SFM), among others, has increased the interest of using point cloud representation in various applications such as digital preservation, surveying, autonomous driving (Chen et al., 2017), 3D gaming, robotics (Oh & Watanabe, 2002), and virtual reality (Park et al., 2008). In return, this new interest has fueled the development of machine learning frameworks that use point clouds as input. Historically, early methods used a preprocessing stage that extracted meticulously hand-crafted features from the point cloud, which were subsequently fed to a neural network (Chen et al., 2003; Rusu et al., 2008; Rusu et al., 2009; Aubry et al., 2011), or they relied on voxelization of the geometry (Savva et al., 2016; Wu et al., 2015; Riegler et al., 2017; Maturana & Scherer, 2015). The PointNet architecture (Qi et al., 2017) was the first to handle raw point cloud data directly and learn features on the fly. This work has spawned several related approaches, aiming to attenuate drawbacks from the original methodology, such as PointNet++ (Qi et al., 2017), or to increase the accuracy and range of application (Wang et al., 2019; Zhai et al., 2020; Li et al., 2018; Liu et al., 2019).

Even though such methods have been quite successful for machine learning problems, they rely on an assumption of locality, which may produce large errors when the underlying task at hand exhibits long-range interactions (LRIs). To capture such interactions using standard convolutional layers, one can use wider window sizes, deeper networks, and/or a large number of features, which may increase the computational cost significantly. Several approaches have been proposed to efficiently capture such interactions in tasks such as semantic segmentation, of which the ideas we briefly summarize below. In the multi-scale type of approaches, features are progressively processed and merged. Within this family, there exist several variants, where the underlying neural networks can

be either recursive neural networks (Ye et al., 2018), convolutional layers (Xu et al., 2019; Xu et al., 2018) or autoencoders (Yang et al., 2018; Deng et al., 2018). Some works have proposed skip connections, following an U-net (Ronneberger et al., 2015) type architecture (Zhou & Tuzel, 2018; Qi et al., 2017), while others have focused on using a tree structure for the clustering of the points (Klokov & Lempitsky, 2017; Zeng & Gevers, 2018; Gadelha et al., 2018), or using an reference permutohedral lattices to compute convolutions (Jampani et al., 2016) whose results are interpolated back to the point cloud (Su et al., 2018). Although these methods have been shown to be successful in a range of applications, when the task at hand presents symmetries, such as rotation, translation, and permutation invariance, there is no systematic framework to embed those symmetries into the algorithmic pipelines. Another line of work, relies on interpreting the point cloud as a graph and use spectral convolutions (Bruna et al.; Defferrard et al., 2016), whose cost can scale super-linearly when dealing with LRIs.

In applications of machine learning to scientific computing, several classical multilevel matrix factorizations have been rewritten in the context of machine learning (Kondor et al., 2014), which have been adapted to handle long-range interactions in the context of end-to-end maps using voxelized geometries in (Fan et al., 2019b;a; Khoo & Ying, 2019; Fan & Ying, 2019) resulting in architectures similar to U-nets (Ronneberger et al., 2015), which have been extended to point clouds in (Li et al., 2020). Due to underlying voxelization of the geometry, it may be difficult for these networks to generalize when the resolution of the voxelization changes.

The efficient treatment of LRI for point clouds is also a prominent problem in many physical applications such as molecular modeling and molecular dynamics simulation. While long-range electrostatic interactions are omnipresent, it has been found that effectively short-ranged models can already describe the $N$-body potential and the associated force field (Behler & Parrinello, 2007; Zhang et al., 2018a;b) for a wide range of physical systems. There have also been a number of recent works aiming at more general systems beyond this regime of effective short-range interactions, such as the work of Ceriotti and co-workers (Grisafi & Ceriotti, 2019; Grisafi et al.; Nigam et al., 2020; Rossi et al., 2020), as well as the works of (Yao et al., 2018; Ko et al., 2009; Hirn et al., 2017; Rupp et al., 2012; Huo & Rupp; Deng et al., 2019; Bereau et al., 2018; Zhang et al., 2019). The general strategy is to build parameterized long-range interactions into the kernel methods or neural network models, so that the resulting model can characterize both short-range, as well as long-range electrostatic interactions. In the neural network context, the computational cost of treating the LRIs using these methods can grow superlinearly with the system size.

The idea of this work is aligned with the approaches in the molecular modeling community, which constructs a neural network layer to directly describe the LRI. In particular, we present a new *long-range convolutional* (LRC)-layer, which performs a global convolutional operation in nearly-linear time with respect to number of units in the layer. By leveraging the non-uniform Fourier transform (NUFFT) (Dutt & Rokhlin, 1993; Greengard & Lee, 2004; Barnett et al., 2019) technique, the LRC-layer implements a convolution with a point-wise multiplication in the frequency domain with trainable weights known as *Fourier multipliers*. The NUFFT is based on the regular fast Fourier transform (FFT) (Cooley & Tukey, 1965) with a fast gridding algorithms, to allow for fast convolution on unstructured data. This new LRC-layer provides a new set of descriptors that can seamlessly satisfy relevant symmetries. For instance, when the kernel of the LRI is rotationally invariant, such symmetry can be directly built into the parameterization of the Fourier kernel. Such descriptors can be used in tandem with the descriptors provided by short-range convolutional layers to improve the performance of the neural network.

Efficient training of a neural network with the LRC-layer for capturing the information of LRIs is another challenging problem. Short-range models can often be trained with data generated with a relatively small computational box (called the small-scale data), and they can be seamlessly deployed in large-scale systems without significantly increasing the generalization error. On the other hand, long-range models need to be trained directly with data generated in a large computational box (called the large-scale data), and the generation process of such large-scale data can be very expensive. For instance, in molecular modeling, the training data is often generated with highly accurate quantum mechanical methods, of which the cost can scale steeply as $\mathcal{O}(N^\alpha)$, where $N$ is the system size and $\alpha \geq 3$. Therefore it is desirable to minimize the number of samples with a large system size. In many applications, the error of the effective short-range model is already modestly small. This motivates us to propose a *two-scale training strategy* as follows. We first generate many small-scale data (cheaply

and possibly in parallel), and train the network without the LRC-layer. Then we use a small number of large-scale data, and perform training with both the short- and long-range convolutional layers.

In order to demonstrate the effectiveness of the LRC-layer and the two-scale training procedure, we apply our method to evaluate the energy and force associated with a model $N$-body potential that exhibit tunable short- and long-range interactions in one, two and three dimensions. The input point cloud consists of the atomic positions, and the output data include the $N$-body potential, local potential, and the force (derivative of the $N$-body potential with respect to atomic positions). In particular, the local potential and the force can be viewed as point clouds associated with the atomic positions. The evaluation of the $N$-body potential is a foundational component in molecular modeling, and LRI plays an important role in the description of ionic systems, macroscopically polarized interfaces, electrode surfaces, and many other problems in nanosciences (French et al., 2010). Our result verifies that the computational cost of the long-range layer can be reduced from $\mathcal{O}(N^2)$ using a direct implementation, to $\mathcal{O}(N)$ (up to logarithmic factors) using NUFFT. Furthermore, we demonstrate that the force, i.e. the derivatives of the potential with respect to *all* inputs can be evaluated with $\mathcal{O}(N)$ cost (up to logarithmic factors). In terms of sample efficiency, we find that for the model problem under study here, the two-scale training strategy can effectively reduce the number of large-scale samples by over an order of magnitude to reach the target accuracy. This can be particularly valuable in the context of molecular modeling, where accurate data are often obtained from first principle electronic structure calculations. Such calculations are often very expensive for large scale systems, and the number of large-scale samples is thus limited.

## 2 LONG-RANGE CONVOLUTIONAL LAYER

Convolutional layers are perhaps the most important building-block in machine learning, due to their great success in image processing and computer vision. A convolutional layer convolves the input, usually an array, with a rectangular mask containing the trainable parameters. When the mask can be kept small (for example while extracting localized features), the convolution layer is highly efficient and effective. A different way for computing a convolution is to use the convolutional theorem as follows: (1) compute the Fourier transform of the input, (2) multiply with the Fourier transform of the mask, i.e.m the Fourier multiplier, and (3) inverse Fourier transform back. In this case, the trainable parameters are the DOFs of the Fourier multipliers and the Fourier transforms are computed using the fast Fourier transform (FFT). This alternative approach is particularly attractive for smooth kernels with large support (i.e., smooth long-range interactions) because the computational cost does not increase with the size of the mask. To the best of our knowledge, this direction has not been explored for LRIs and below we detail now to apply this to point clouds.

Given a point cloud $\{x_i\}_{i=1}^N \subset \mathbb{R}^d$ and scalar weights $\{f_i\}_{i=1}^N$, we consider the problem of computing the quantity $u_i := \sum_{j=1}^N \phi_\theta(x_i - x_j) f_j$ at each $x_i$. Here the function $\phi_\theta(\cdot)$ is the kernel with a *generic* trainable parameter $\theta$. At first glance the cost of this operation scales as $\mathcal{O}(N^2)$: we need to evaluate $u_i$ for each point $x_i$, which requires $\mathcal{O}(N)$ work per evaluation. By introducing a generalized function $f(y) = \sum_i f_i \cdot \delta(y - x_i)$ and defining a function $u(x) = \int \phi_\theta(x - y) f(y) dy$, one notices that $u_i$ is the value of $u(x)$ at $x = x_i$. The advantage of this viewpoint is that one can now invoke the connection between convolution and Fourier transform

$$\hat{u}(k) = \hat{\phi}_\theta(k) \cdot \hat{f}(k), \tag{1}$$

where $\hat{\phi}_\theta(k)$ is a trainable Fourier multiplier. This approach is suitable for point clouds since the trainable parameters are decoupled from the geometry of the point cloud. To make this approach practical, one needs to address two issues: (1) the non-uniform distribution of the point cloud and (2) how to represent the multiplier $\hat{\phi}_\theta(k)$.

**Non-uniform distribution of the point cloud** Equation 1 suggests that one can compute the convolution directly using the convolution theorem, which typically relies on the FFT to obtain a low-complexity algorithm. Unfortunately, $\{x_i\}_{i=1}^N$ do not form a regular grid, thus FFT can not be directly used. We overcome this difficulty by invoking the NUFFT[1] (Dutt & Rokhlin, 1993), which serves as the corner-stone of our instance of the LRC-layer[2].

---

[1]See Appendix C.2 for further details.

[2]We point out, that one could in practice use an fast summation algorithm, such as the fast multipole method (FMM) introduced by Greengard & Rokhlin (1987), to evaluate $u_i$. This would results in the same complexity if

---

**Algorithm 1** Long-range convolutional layer

    Input: $\{x_i\}_{i=1}^N$, $\{f_i\}_{i=1}^N$
    Output: $\{x_i\}_{i=1}^N$, $\{u_i\}_{i=1}^N$, where $u_i = \sum_{j=1}^N f_j \phi_\theta(x_i - x_j)$.
1:  Define the generalized function: $f(x) = \sum_{j=1}^N f_j \delta(x - x_j)$
2:  Mollify the Dirac deltas: $f_\tau(x) = \sum_{j=1}^N f_j g_\tau(x - x_j)$, where $g_\tau$ is defined in Appendix C.2
3:  Sample in a regular grid: $f_\tau(x_\ell) = \sum_{j=1}^N g_\tau(x_\ell - x_j)$ for $x_\ell$ in grid of size $L_{\text{FFT}}$ in each dim
4:  Compute FFT: $F_\tau(k) = \texttt{FFT}(f_\tau)(k)$
5:  Re-scale the signal: $F(k) = \sqrt{\frac{\pi}{\tau}} e^{k^2 \tau} F_\tau(k)$
6:  Multiply by Fourier multipliers: $\hat{v}(k) = \hat{\phi}_\theta(k) \cdot F(k)$
7:  Re-scale the signal: $\hat{v}_{-\tau}(k) = \sqrt{\frac{\pi}{\tau}} e^{k^2 \tau} \hat{v}(k)$
8:  Compute IFFT: $u_{-\tau}(x_\ell) = \texttt{IFFT}(\hat{v}_{-\tau})(x)$ for $x_\ell$ on the regular grid
9:  Interpolate to the point cloud: $u_i = u(x_i) = u_{-\tau} * g_\tau(x_i)$

---

The LRC-layer is summarized in Alg. 1, where $\tau$ is chosen following Dutt & Rokhlin (1993). The inputs of this layer are the point cloud $\{x_i\}_{i=1}^N$ and the corresponding weights $\{f_i\}_{i=1}^N$. The outputs are $u_i \equiv u(x_i)$ for $i = 1, ..., N$. The number of elements in the underlying grid $N_{\text{FFT}} = L_{\text{FFT}}^d$ is chosen such that the kernel is adequately sampled and the complexity remains low. As shown in Appendix C.5, one only needs a relatively small $L_{\text{FFT}}$. Even though the precise number is problem-specific, given that the goal is to approximate LRIs that are supposedly smooth, it can be captured with a relatively small number of Fourier modes.

The LRC-layer is composed of three steps: (1) It computes the Fourier transform from the point cloud to a regular grid using the NUFFT algorithm (lines $2 - 5$ in Alg. 1 and showcased in Fig. 2). (2) It multiplies the result by a set of trainable Fourier multipliers (line 6 in Alg. 1). (3) It computes the inverse Fourier transform from the regular grid back to the point cloud (lines $7 - 9$ in Alg. 1).

Within the LRC-layer in Alg. 1, the only trainable component is the parameter $\theta$ of the Fourier multiplier $\hat{\phi}_\theta(k)$. The remaining components, including the mollifier $g_\tau(\cdot)$ and the Cartesian grid size, are taken to be fixed. One can, in principle, train them as well, but it comes with a much higher cost. Among the steps of Alg. 1, the sampling operator, the rescaling operator, the interpolation operator, and the Fourier transforms, are all *linear and non-trainable*. Therefore, derivative computations of backpropagation just go through them directly.

Alg. 1 is presented in terms of only one single channel or feature dimension, i.e., $f_j \in \mathbb{R}$ and $u_i \in \mathbb{R}$. However, it can be easily generalized to multiple channels, for example $f_j \in \mathbb{R}^{d_1}$ and $u_i \in \mathbb{R}^{d_2}$. In this case, the Fourier multiplier $\hat{\phi}_\theta(k)$ at each point $k$ is a $d_2 \times d_1$ matrix, and all Fourier transforms are applied component-wise.

**Representation of the Fourier multiplier** A useful feature of the LRC-layer is that it is quite easy to impose symmetries on the Fourier multipliers. For example, if the convolution kernel $\phi_\theta(\cdot)$ is constrained to have parity symmetry, rotational symmetry, smoothness or decay properties, these constraints can be imposed accordingly on the coefficients of the Fourier multipliers $\hat{\phi}_\theta(k)$. When the size of the training data is limited, it is often necessary to reduce the number of trainable parameters in order to regularize the kernel. For example, we may parameterize the Fourier multiplier as a linear combination of several predetermined functions on the Fourier grid. This is the procedure used in molecular modeling (Grisafi & Ceriotti, 2019; Yao et al., 2018; Ko et al., 2009), and also in our numerical examples in equation 7. We also remark that the LRC-layer described here can be applied to point clouds a way similar to a standard convolution layer applied to images and multiple LRC-layers can be composed on top of each other.

---

the kernel is fixed. However, in order for the kernel to be trainable, this would require a different algorithm for each iteration, including the computation of the derivatives, thus increasing the computational cost and rendering the implementation significantly more cumbersome.

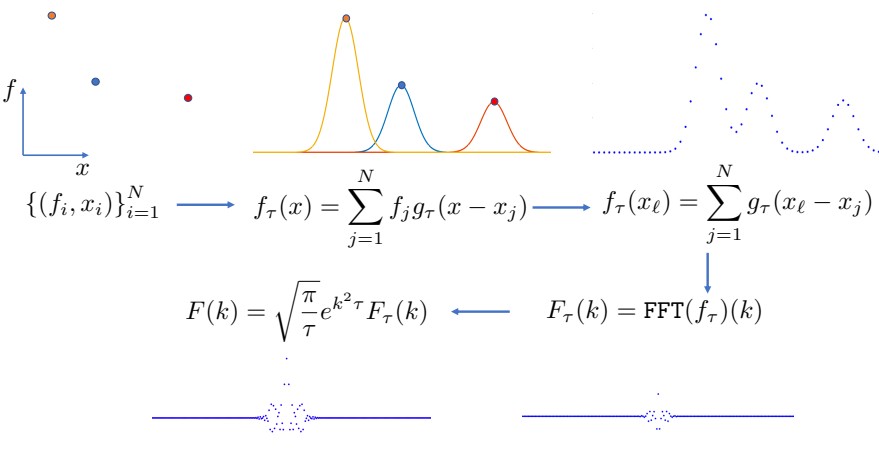

Figure 1: Diagram of the NUFFT. Starting from the cloud point $\{x_i\}_{i=1}^N$, we form the mollified function $f_\tau$, sample it in a regular grid, compute the Fourier transform $F_\tau(k)$ of the sampled function. Finally in order to obtain $F(k)$, we rescale the signal to undo the spatial convolution.

## 3  LEARNING THE $N$-BODY POTENTIAL

To demonstrate the effectiveness of the LRC-layer, we consider the problem of learning the energy and force associated with a model $N$-body potential in the context of molecular modeling. As mentioned in Section 1, the potential evaluation often invokes expensive ab-initio calculations that one would like to bypass for efficiency reasons.

The setup of this learning problem is as follows. First, we assume access to a *black-box* model potential, which consists of both short- and long-range interactions. However, internal parameters of the potential are inaccessible to the training architecture and algorithm. A set of training samples are generated by the model, where each sample consists of a configuration of the points $\{x_i\}$ along with the potential and force. Second, we set up a deep neural network that includes (among other components) the LRC-layer for addressing the long-range interaction. This network is trained with stochastic gradient type of algorithms using the collected dataset and the trained network can be used for predicting the potential and forces for new point cloud configurations. These two components are described in the following two subsections in detail.

### 3.1  MODEL PROBLEM AND DATA GENERATION

**Model**  We suppose that $\Omega = [0, L]^d$, and we denote the point cloud by $\mathbf{x} = \{x_i\}_{i=1}^N \subset \Omega \subset \mathbb{R}^d$, for $d = 1, 2$, or 3. We define the total energy, the local potential and the forces acting on particle $j$ by

$$U = \sum_{1 \leq i < j \leq N} \psi(x_i - x_j), \qquad U_j(x) = \sum_{i \neq j} \psi(x_i - x), \qquad \text{and} \qquad F_j = -\partial_x U_j(x)|_{x=x_j}, \quad (2)$$

respectively, where the interaction kernel $\psi(r)$ is a smooth function, besides a possible singularity at the origin and decreases as $\|r\| \to \infty$.

**Sampling**  We define a snapshot as one *configuration*[3] of particles, $\mathbf{x}^\ell = \{x_j^{[\ell]}\}_{j=1}^N$, together with the global energy $U^{[\ell]}$ and the forces $F^{[\ell]}$, where $\ell$ is the index representing the number in the training/testing set. We sample the configuration of particles $\mathbf{x}^\ell$ randomly, with the restriction that two particles can not be closer than a predetermined value $\delta_{\texttt{min}}$ in order to avoid the singularity. After an admissible configuration is computed we generate the energy and forces following Appendix B. This process is repeated until obtaining $N_{\texttt{sample}}$ snapshots.

---

[3]For the sake of clarity, we suppose that the number of particles at each configuration is the same.

## 3.2 ARCHITECTURE

Our network architecture consists of separate descriptors for the short- interactions and long-range interactions, respectively. To capture the short-range interaction, we compute a local convolution using for each point only its neighboring points within a ball of predetermined radius. For the long-range interactions, we compute an all-to-all convolution using the LRC-layer introduced in Section 2, whose output is distributed to each particle and then fed to a sequence of subsequent layers.

**Short-range descriptor** For a given particle $x_i$, and an interaction radius $R$, we define $\mathcal{I}_i$, the interaction list of $x_i$, as the indices $j$ such that $\|x_i - x_j\| < R$, i.e., the indices of the particles that are inside a ball of radius $R$ centered at $x_i$. Thus for each particle $x_i$ we build the generalized coordinates $s_{i,j} = x_i - x_j$, and the short-range descriptor

$$\mathcal{D}_{\mathrm{sr}}^i = \sum_{j \in \mathcal{I}_i} f_\theta(s_{i,j}), \tag{3}$$

where $f_\theta : \mathbb{R}^d \to \mathbb{R}^{m_{\mathrm{sr}}}$ is a function represented by a neural network specified in Appendix C.1, where $m_{\mathrm{sr}}$ is the number of short-range features. By construction $f_\theta(s)$ is smooth and it satisfies $f_\theta(s) = 0$ for $\|s\| > R$.

**Long-range descriptor** We feed the LRC-layer with the raw point cloud represented by $\{x_i\}_{i=1}^N$ with weights $\{f_i\}_{i=1}^N$, which for simplicity can be assumed to be equal to one here, i.e., $f_i = 1$ for $i = 1, ..., N$. The output of the layer is a two-dimensional tensor $u^k(x_i)$ with $i = 1, \ldots, N$ and $k = 1, \ldots, K_{\mathrm{chnls}}$. Then for each $x_i$, its corresponding slice given by the vector $[u^1(x_i), u^2(x_i), \cdots, u^{K_{\mathrm{chnls}}}(x_i)]$, is fed to a function $g_\theta : \mathbb{R}^{K_{\mathrm{chnls}}} \to \mathbb{R}^{m_{\mathrm{lr}}}$, which is represented by a neural network with non-linear activation functions. Here $\theta$ is a generic set of trainable parameters and $m_{\mathrm{lr}}$ is the number of long-range features. The descriptor for particle $x_i$, which depends on all the other particles thanks to the LRC-layer, is defined by

$$\mathcal{D}_{\mathrm{lr}}^i = g_\theta(u^1(x_i), u^2(x_i), \cdots, u^{K_{\mathrm{chnls}}}(x_i)) \tag{4}$$

**Short-range network** When only the short-range interaction is present, the short-range descriptor for each particle is fed particle-wise to a fitting network $\mathcal{F}_{\mathrm{sr}} : \mathbb{R}^{m_{\mathrm{sr}}} \to \mathbb{R}$. In this case $\mathcal{F}_{\mathrm{sr}}(\mathcal{D}_{\mathrm{sr}}^i)$ only depends on particle $x_i$ and its neighbors. Finally, the contributions from each particle are accumulated so the short-range neural network (NN) energy and forces are given by

$$U_{\mathrm{sr}}^{\mathrm{NN}} = \sum_{i=1}^N \mathcal{F}_{\mathrm{sr}}(\mathcal{D}_{\mathrm{sr}}^i) \qquad \text{and} \qquad (F_{\mathrm{sr}}^{\mathrm{NN}})_j = -\partial_{x_j} U_{\mathrm{sr}}^{\mathrm{NN}} \tag{5}$$

respectively (see Fig. 2(left)). The derivatives are computed using Tensorflow (Abadi et al., 2015) directly. This network as shown by (Zhang et al., 2018b) is rotation, translation, and permutation invariant (Zaheer et al., 2017).

We point out that this architecture can be understood as a non-linear local convolution: for each particle $i$ one applies the same function $f_\theta$ to each of its neighbors. The result is then pooled into the descriptor $\mathcal{D}_{\mathrm{sr}}^i$, then processed locally by $\mathcal{F}_{\mathrm{sr}}$ (akin to a non-linear convolution with a filter of width one), and finally pooled globally into $U_{\mathrm{sr}}^{\mathrm{NN}}$.

**Full-range network** When both the short-range and long-range interactions are present, the long range descriptor and the local descriptor are combined and fed particle-wise to a fitting network $\mathcal{F} : \mathbb{R}^{m_{\mathrm{sr}} + m_{\mathrm{lr}}} \to \mathbb{R}$ to produce the overall neural network (NN) energy and forces

$$U^{\mathrm{NN}} = \sum_{i=1}^N \mathcal{F}(\mathcal{D}_{\mathrm{sr}}^i, \mathcal{D}_{\mathrm{lr}}^i), \qquad \text{and} \qquad (F^{\mathrm{NN}})_j = -\partial_{x_j} U^{\mathrm{NN}} \tag{6}$$

respectively (see Fig. 2(right)). Following Section 2, the long-range descriptor is translation invariant by design and can be easily made rotation invariant. Furthermore, it is well known (Zaheer et al., 2017) that this construction is permutation invariant. Further details on the implementation of the network can be found in Appendix C.3. From the structures shown in Fig. 2[4], it is clear that we can recover the first architecture from the second, by zeroing some entries at the fitting network, and removing the LRC-layer.

---

[4]We provide more detailed schematics in Fig. 6 and Fig. 7 in Appendix C.1

Figure 2: (left) The short-range network architecture. (right) The full-range network architecture.

Finally, let us comment on the inference complexity of the proposed network where, for simplicity we assume that $\mathcal{O}(K_{\text{chnls}}) = \mathcal{O}(m_{\text{sr}}) = \mathcal{O}(m_{\text{lr}}) = \mathcal{O}(1)$, and that the depth of the neural networks is $\mathcal{O}(1)$. The cost for computing $U_{\text{sr}}^{\text{NN}}$ is $\mathcal{O}(N)$, provided that each particle has a bounded number of neighbors. The complexity for computing the forces also scales linearly in $N$, albeit with higher constants. The complexity of computing both $U^{\text{NN}}$ and associated forces[5] is $\mathcal{O}(N + N_{\text{FFT}} \log N_{\text{FFT}})$.

## 4 NUMERICAL RESULTS

The loss function is the mean squared error of the forces $\frac{1}{N_{\text{sample}}} \sum_{\ell=1}^{N_{\text{sample}}} \sum_{i=1}^{N} \left\| F_\theta^{\text{NN}}(x_i^{[\ell]}) - F_i^{[\ell]} \right\|^2$, where the $i$-index runs on the points of each snapshot, and $\ell$ runs on the test samples. We also generate 100 snapshots of data to test the performance of network. This particular loss could lead to shift the potential energy by up to a global constant, which can be subsequently fixed by including the error of the energy in the loss (Zhang et al., 2018b). For the testing stage of we use the relative $\ell^2$ error of the forces as metric, which is defined as $\epsilon_{\text{rel}} := \sqrt{\sum_{\ell,i} \|F_i^{[\ell]} - F_\theta^{\text{NN}}(x_i^{[\ell]})\|^2 / \sum_{\ell,i} \|F_i^{[\ell]}\|^2}$. The standard training parameters are listed in Appendix C.4.

The experiments shown in the sequel are designed to provide a fair comparison with state-of-the-art methods for localized interactions. They showcase that, by adding a single LRC-layer, one can outperform these methods significantly.

The kernels $\psi$ used in the experiment typically exhibit two interaction lengths: $\psi(\cdot) \equiv \alpha_1 \psi^{\mu_1}(\cdot) + \alpha_2 \psi^{\mu_2}(\cdot)$, where each of $\psi^{\mu_1}$ and $\psi^{\mu_2}$ is either a simple exponential kernel or screened-Coulomb kernel (also known as the Yukawa kernel). For each of $\psi^{\mu_1}$ and $\psi^{\mu_2}$, the superscripts denote the reciprocal of the interaction length, i.e., length scale $\sim \mu_1^{-1}$ or $\sim \mu_2^{-1}$. Without loss of generality, $\mu_1 > \mu_2$, so that $\mu_1$ corresponds to the short-range scale and $\mu_2$ the long-range scale. We also assume that $0 \leq \alpha_2 \leq \alpha_1$ and $\alpha_1 + \alpha_2 = 1$, so that the effect of the long-range interaction can be smaller in magnitude compared to that of the short-range interaction. In the special case of $\alpha_2 = 0$, the kernel exhibits only a single scale $\sim \mu_1^{-1}$. The precise definition of the kernel depends on the spatial dimension and boundary conditions, which are explained in Appendix B.

For a fixed set of kernel parameters $(\mu_1, \mu_2, \alpha_1, \alpha_2)$, we consider two types of data: large- and small-scale data, generated in the domains $\Omega_{\text{lr}}$ and $\Omega_{\text{sr}}$ respectively (details to be defined in each experiment).

The Fourier multiplier within the LRC-layer is parameterized as

$$\hat{\phi}_{\beta,\lambda}(k) = \frac{4\pi\beta}{|k|^2 + \lambda^2}, \tag{7}$$

where $\beta$ and $\lambda$ are trainable parameters. This is a simple parameterization, and a more complex model can be used as well with minimal changes to the procedure. For all experiments shown below, two kernel channels are used and as a result there are only four trainable parameters in the LRC-layer.

The numerical results aim to show namely two properties: i) the LRC-layer is able to efficiently capture LRIs, and ii) the two-scale training strategy can reduce the amount of large-scale data significantly. To demonstrate the first property, we gradually increase the interaction length of the kernel. The accuracy of the short-range network with a fixed interaction radius is supposed to decrease rapidly, while using the LRC-layer improves the accuracy significantly. To show the second property, we generate data with two interaction lengths and train the full-range network using the one- and

---

[5]See Appendix C.2 and C.3 for further details.

Table 1: Relative testing error for trained screened-Coulomb type 1D models with $\alpha_1 = 1$, $\alpha_2 = 0$, and varying $\mu_1$. Notice that $\mu_2$ can be arbitrary here given that $\alpha_2 = 0$.

| $\mu_1$ | 0.5 | 1.0 | 2.0 | 5.0 | 10.0 |
|---|---|---|---|---|---|
| short-range network | 0.05119 | 0.02919 | 0.00597 | 0.00079 | **0.00032** |
| full-range network | **0.00828** | **0.00602** | **0.00336** | **0.00077** | 0.00054 |

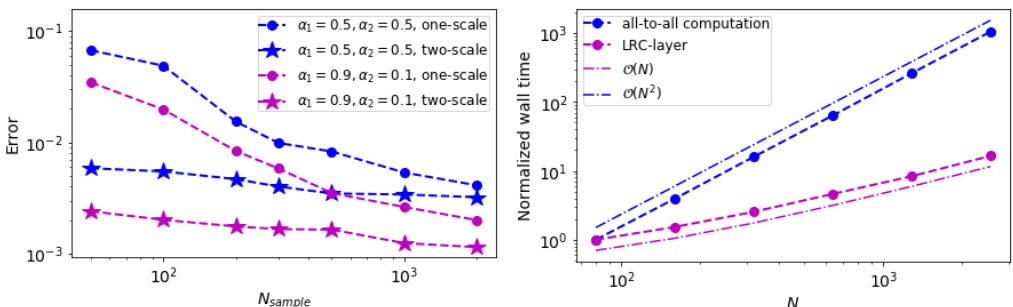

Figure 3: (left) Testing error of the trained 1D model with respect to the number of snapshots using the one- and two-scale training strategies using data generated with the screened-Coulomb potential and parameters $\mu_1 = 5.0$, $\mu_2 = 0.5$ (right) normalized wall-time for the LRC and the direct all-to-all computation.

two-scale strategies. Finally, we also aim to demonstrate that the LRC-layer is competitive against a direct convolution in which the all-to-all computation is performed explicitly.

**1D** In the first set of experiments, the domain $\Omega = [0, 5]$, $N = 20$ and $N_{\texttt{sample}} = 1000$, where $N_{\texttt{sample}}$ is the number of snapshots and $N$ is the total number of points in each snapshot. For the kernel, we set $\alpha_2$ and vary $\mu_1$ to generate datasets at different interaction lengths. For each dataset we train both short-range and full-range networks using the one-scale data. The results are summarized in Table 1, where we can observe that as the characteristic interaction length increases, the accuracy of the short-range network decreases while using the full-range network can restore the accuracy. This experiment shows that local networks are often highly accurate when the interactions are localized, but the accuracy quickly deteriorates as the interaction length increases (i.e. as $\mu_1$ decreases).

For the second set of experiments we used two sets of kernel parameters: one heavily biased towards a localized interaction length, and another in which both interaction lengths are equally weighted. For each set of kernel parameters, we generate $10,000$ small-scale snapshots using $\Omega_{\texttt{sr}} = [0, 5]$ and $N = 20$, and a large number of large-scale snapshots using $\Omega_{\texttt{lr}} = [0, 50]$ and $N = 200$ particles. The interaction radius $R = 1.5$, $\delta_{\texttt{min}} = 0.05$, and $N_{\texttt{FFT}}$ is 501. We train the network with the one- and two-scale training strategies described in the prequel. Fig. 3 (left) depicts the advantage of using the two-scale training strategy: we obtain roughly the same accuracy at a fraction of the number of large-scale training samples. We observe that when the number of large-scale training samples is sufficiently large, the resulting test accuracy is independent of the training strategy. We also observe that the training dynamics is stable with respect to different random seeds.

We compare the LRC-layer with a direct all-to-all computation. We benchmark the wall time of both layers, with increasingly number of particles. To account for implementation effects we normalize the wall times in Fig. 3 (right) and the results corroborate the complexity claims made in Section 2.

**2D** We perform the same experiments as in the one-dimensional case. We fix $\Omega = [0, 15]^2$, $N = 450$ and $N_{\texttt{sample}} = 10000$. The results are summarized in Table 2, which shows that as $\mu$ decreases, the full-range network outperforms the short-range one.

For the second set of experiments, $R = 1.5$, $\delta_{\texttt{min}} = 0.05$, and $N_{\texttt{FFT}}$ is $31^2$. For the small-scale data, $\Omega_{\texttt{sr}} = [0, 3]^2$, $N = 18$, and $N_{\texttt{sample}} = 10,000$. For the large-scale data, $\Omega_{\texttt{lr}} = [0, 15]^2$, $N = 450$. Similarly to the 1D case, we train the networks with both strategies using different amounts of large-scale data. The results summarized in Fig. 4 show that the two-scale strategy efficiently captures the long-range interactions with only a small number of the long-range training samples.

Table 2: Relative testing error for trained screened-Coulomb type 2D models with $\alpha_1 = 1, \alpha_2 = 0$, and varying $\mu_1$. Again $\mu_2$ can be arbitrary given that $\alpha_2 = 0$.

| $\mu_1$ | 1.0 | 2.0 | 5.0 | 10.0 |
|---|---|---|---|---|
| short-range network | 0.07847 | 0.02332 | 0.00433 | 0.00242 |
| full-range network | **0.00785** | **0.00526** | **0.00363** | **0.00181** |

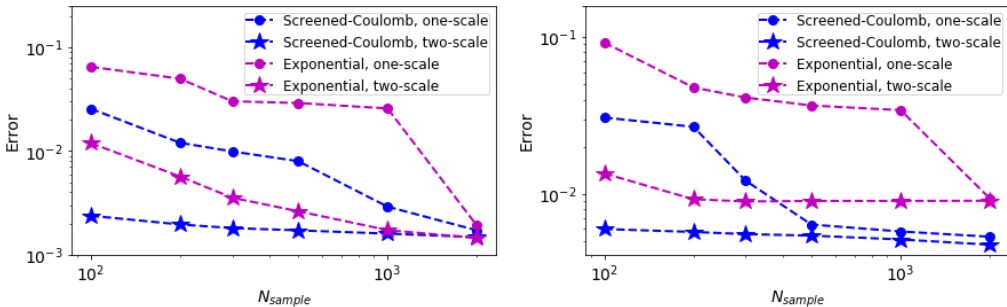

Figure 4: Testing error of the trained 2D model with respect to the number of snapshots using the one- and two-scale training strategies using both screened-Coulomb and exponential potentials with $\mu_1 = 10$, $\mu_2 = 1$ : (left) $\alpha_1 = 0.9$, and $\alpha_2 = 0.1$; and (right) $\alpha_1 = 0.5$, and $\alpha_2 = 0.5$.

Analogously to the 1D case, we can observe that for sufficiently large-scale training samples the resulting test accuracy is identical regardless of the training strategy used. Also similar to Fig. 3 (left), we find that the lowest achievable test error is larger in Fig. 4 (right, with a larger $\alpha_2$) than that in Fig. 4 (left, with a smaller $\alpha_2$). Nonetheless, we observe that the test error of the two-scale training strategy becomes less sensitive with respect to the number of training samples when $\alpha_2$ becomes larger, i.e. the LRI becomes more prominent.

**3D** The domain $\Omega$ is $[0, 3]^3$ with 2 points in each of the 27 unit cells. The other parameters are the interaction radius $R = 1.0$, $\delta_{\min} = 0.1$, and $N_{\texttt{sample}} = 1000$. The Fourier domain used is of size $N_{\text{FFT}} = 25^3$. The results in Table 3 demonstrate that full-range network is capable of maintaining good accuracy for a wide range of characteristic interactions lengths.

## 5 CONCLUSION

We have presented an efficient long-range convolutional (LRC) layer, which leverages the non-uniform fast Fourier transform (NUFFT) to reduce the cost from quadratic to nearly-linear with respect to the number of degrees of freedom. We have also introduced a two-scale training strategy to effectively reduce the number of large-scale samples. This can be particularly important when the generation of these large-scale samples dominates the computational cost. While this paper demonstrates the effectiveness of the LRC-layer for computing the energy and force associated with a model $N$-body potential, we expect the LRC-layer to become a useful component in designing neural networks for modeling real chemical and materials systems, where the LRI cannot be accurately captured using short ranged models. We also expect that the LRC-layer can be a useful tool for a wide range of machine learning (such as regression and classification) tasks.

Table 3: Relative testing error for trained exponential type 3D models with $\alpha_1 = 1, \alpha_2 = 0$, and varying $\mu_1$. Again $\mu_2$ can be arbitrary given that $\alpha_2 = 0$.

| $\mu_1$ | 5 | 7.5 | 10 |
|---|---|---|---|
| short-range network | 0.06249 | 0.01125 | 0.00175 |
| full-range network | **0.00971** | **0.00411** | **0.00151** |

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

## A   NOTATION

A table of notations is summarized in Table 4.

## B   DATA GENERATION

We provide further details about the data generation process and how the parameter $\mu$ dictates the characteristic interaction length.

**Exponential kernel:**  Suppose $\Omega$ be the torus $[0, L]^d$ and that $\mathbf{x} = \{x_i\}_{i=1}^N \subset \Omega \subset \mathbb{R}^d$ for $d = 1, 2$, or $3$. The exponential kernel is defined as

$$\psi^\mu(x - y) = e^{-\mu\|x-y\|}, \tag{8}$$

where $\|\cdot\|$ is the Euclidean norm over the torus. Following Section 3.1 we define the total energy and the potential as

$$U = \sum_{i<j}^N e^{-\mu\|x_i-x_j\|} \qquad \text{and} \qquad U_j(x) = \sum_{i\neq j}^N e^{-\mu\|x_i-x\|}, \tag{9}$$

respectively. The forces are given by

$$F_j = -\partial_{x_j} U_j(x_j) = -\sum_{i\neq j}^N \frac{x_i - x_j}{\|x_i - x_j\|} \mu e^{-\mu\|x_i-x_j\|}. \tag{10}$$

**Screened-Coulomb kernel:** In 3D, the screened-Coulomb potential with free space boundary condition is given by

$$\psi^\mu(x - y) = \frac{1}{4\pi\|x - y\|} e^{-\mu\|x-y\|}. \tag{11}$$

Table 4: Symbols introduced in the current paper with their corresponding meaning.

| Symbol | Meaning |
|--------|---------|
| **Notation** | |
| **Data** | |
| $d$ | Spatial dimension of the problem |
| $\Omega = [0, L]^d \subset \mathbb{R}^d$ | Computational Domain |
| $\{x_i\}_{i=1}^N \subset \Omega$ | Point cloud |
| $N$ | Number of points in the point cloud |
| $N_{\texttt{sample}}$ | Number of snapshots for training |
| $\psi^\mu$ | Interaction kernel |
| $\mu$ | Inverse characteristic interaction length |
| $U$ | Potential |
| $F_j$ | Forces exerted over the $j$-th particle |
| **Networks** | |
| $\mathcal{D}^i$ | Descriptor associated to $x_i$ |
| $\mathcal{F}$ | Fitting Network |
| $\theta$ | Generic trainable parameters |
| $f_\theta$ | Trainable function inside the descriptor |
| $g_\theta$ | Trainable function inside the fitting network |
| $R$ | Interaction radius |
| **LRC-layer** | |
| $g_\tau$ | Mollifier of the Dirac deltas defined in equation 20 |
| $\tau$ | Broadening factor in the mollifier |
| FFT, IFFT | Fast Fourier transform and its inverse |
| $\phi_\theta$ | Kernel with trainable parameters $\theta$ |
| $\hat{\phi}_\theta$ | Fourier transform of the kernel |
| $L_{\texttt{FFT}}$ | Number of Fourier modes per dimension |
| $N_{\texttt{FFT}} = L_{\texttt{FFT}}^d.$ | Total number of Fourier modes |

Over the torus $[0, L]^d$, the kernel $\psi^\mu(x - y)$ is the Green's function $G^\mu(x, y)$ defined via

$$\Delta G^\mu(x, y) - \mu^2 G^\mu(x, y) = -\delta_y(x), \tag{12}$$

with the periodic boundary condition. In order to compute the screened-Coulomb potential numerically, a spectral method is used: in particular,

$$\psi^\mu(x - y) = G^\mu(x, y) = \mathcal{F}^{-1}\left(\frac{e^{ik \cdot y}}{\|k\|^2 + \mu^2} \chi_\epsilon(k)\right), \tag{13}$$

where $\mathcal{F}^{-1}$ stands for the inverse Fourier transform and $\chi_\epsilon(k)$ is a smoothing factor, usually Gaussian, to numerically avoid the Gibbs phenomenon. Similar to the exponential case, the parameter $\mu$ controls the localization of the potential. In addition, the derivatives are taken numerically in the Fourier domain.

**Visualization:** To visualize the relation between $\mu$ and the characteristic interaction length in 1D, consider a given particle, e.g., $x_{100}$ and compute the force contribution from the other particles. Fig. 5 shows that force contribution is extremely small outside a small interaction region for $\mu = 5.0$ while the interaction region for $\mu = 0.5$ is much larger.

## C    DETAILS OF ARCHITECTURE AND TRAINING

### C.1    SHORT-RANGE DESCRIPTOR

Here we specify the structure of $\mathcal{D}^i$ introduced in Section 3.2. For a given particle $x_i$, and an interaction radius $R$, define the interaction list $\mathcal{I}_i$ of $x_i$ as the set of indices $j$ such that $\|x_i - x_j\| < R$, where $\|\cdot\|$ stands for the distance over the torus $[0, L]^d$. To simplify the discussion, we assume that there exists a maximal number of neighbors $N_{\texttt{maxNeigh}}$ for each $x_i$. We stack the neighbors in a tensor whose dimensions are constant across different particles. This value is chosen to be sufficiently

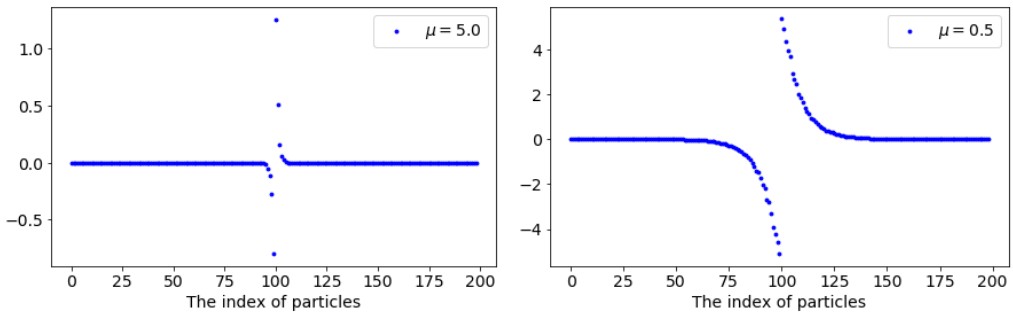

Figure 5: The force contribution to particle $x_{100}$ from other particles. Results are shown for two different characteristic interaction lengths.

large to cover the number of elements in the interaction list. If the cardinality of $\mathcal{I}_i$ is less than $N_{\texttt{maxNeigh}}$, we pad the tensor with dummy values.

In the 1D case the generalized coordinates are defined as

$$s_{i,j} = \|x_i - x_j\|, \qquad \text{and} \qquad r_{i,j} = \frac{1}{\|x_i - x_j\|} \qquad (14)$$

for $j \in \mathcal{I}_i$. We introduce two fully-connected neural networks $f_{\theta_1}, f_{\theta_2} \colon R^+ \to \mathbb{R}^{m_{\text{sr}}/2}$, where each consists of five layers with the number of units doubling at each layer and ranging from 2 to 32. The activation function after each layer is $\tanh$ and the initialization follows Glorot normal distribution.

For particle $x_i$ the short-range descriptor is defined as the concatenation of

$$\mathcal{D}^i_{1,\text{sr}} = \sum_{j \in \mathcal{I}_i} f_{\theta_1}(\hat{s}_{i,j})\hat{r}_{i,j} \qquad \text{and} \qquad \mathcal{D}^i_{2,\text{sr}} = \sum_{j \in \mathcal{I}_i} f_{\theta_2}(\hat{r}_{i,j})\hat{r}_{i,j}, \qquad (15)$$

where $\hat{r}_{i,j}, \hat{s}_{i,j}$ are the normalized copies of $r_{i,j}$ and $s_{i,j}$ with mean zero and standard deviation equals to one. The mean and standard deviation are estimated by using a small number of snapshots. We multiply the network's output $f_\theta$ by $\hat{r}_{i,j}$ (which is zero if $j$ is a dummy particle). This procedure enforces a zero output for particles not in the interaction list. The construction satisfies the design requirement mentioned in Section 3.2.

In the short-range network, one concatenates the two descriptor above and feeds them particle-wise to the short-range fitting network. The fitting network $\mathcal{F}_{\text{sr}} \colon \mathbb{R}^{m_{\text{sr}}} \to \mathbb{R}$ is a residual neural network (ResNet) with six layers, each with 32 units. The activation function and initialization strategy are the same as the ones for the short-range descriptors. Fig. 6 shows the detailed architecture of the short-range network.

$$U^{\text{NN}}_{\text{sr}} = \sum_{i=1}^N \mathcal{F}(\mathcal{D}^i_{\text{sr}}) = \sum_{i=1}^N \mathcal{F}(\mathcal{D}^i_{1,\text{sr}}, \mathcal{D}^i_{2,\text{sr}}) \qquad (16)$$

In 2D and 3D, there is a slight difference of generalized coordinates: we compute

$$s_{i,j} = \frac{x_i - x_j}{\|x_i - x_j\|} \qquad \text{and} \qquad r_{i,j} = \frac{1}{\|x_i - x_j\|}, \qquad (17)$$

where $s_{i,j}$ is a vector now. The local descriptors are defined in the following forms:

$$\mathcal{D}^i_{1,\text{sr}} = \sum_{j \in \mathcal{I}_i} f_{\theta_1}(s_{i,j})\hat{r}_{i,j} \qquad \text{and} \qquad \mathcal{D}^i_{2,\text{sr}} = \sum_{j \in \mathcal{I}_i} f_{\theta_2}(\hat{r}_{i,j})\hat{r}_{i,j} \qquad (18)$$

## C.2 NUFFT

In this section we provide further details for the NUFFT implementation. Suppose that the input of the NUFFT is given by $\{x_i\}_{i=1}^N \subset \mathbb{R}^d$, where each point has a given associated weight $f_i$. The first

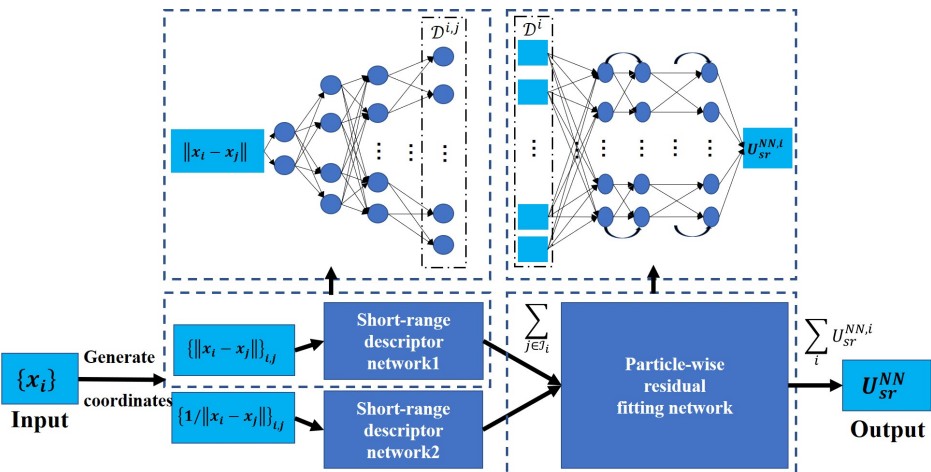

Figure 6: The structure of short-range network for 1D case.

step is to construct the weighted train of Dirac deltas as

$$f(x) = \sum_{j=1}^{N} f_j \delta(x - x_j). \tag{19}$$

We point out that in some of the experiments $f_j$ simply equals to 1. One then defines a periodic Gaussian convolution kernel

$$g_\tau(x) = \sum_{\ell \in \mathbb{Z}^d} e^{-\|x - \ell L\|^2 / 4\tau}, \tag{20}$$

where $L$ is the length of the interval and $\tau$ determines the size of mollification. In practice a good choice is $\tau = 12\left(\frac{L}{2\pi L_{\text{FFT}}}\right)^2$ (Dutt & Rokhlin, 1993), where $L_{\text{FFT}}$ is the number of points in each dimension and $N_{\text{FFT}} = L_{\text{FFT}}^d$. We define

$$f_\tau(x) = f * g_\tau(x) = \int_{[0,L]^d} f(y) g_\tau(x - y) dy = \sum_{j=1}^{N} f_j g_\tau(x - x_j). \tag{21}$$

With the Fourier transform defined as

$$F_\tau(k) = \frac{1}{L^d} \int_{[0,L]^d} f_\tau(x) e^{-i2\pi k \cdot x / L} dx \tag{22}$$

for $k \in \mathbb{Z}^d$, we compute its discrete counterpart

$$F_\tau(k) \approx \frac{1}{N_{\text{FFT}}} \sum_{m \in [0, L_{\text{FFT}} - 1]^d} f_\tau(Lm/L_{\text{FFT}}) e^{-i2\pi k \cdot m / L_{\text{FFT}}} \tag{23}$$

$$\approx \frac{1}{N_{\text{FFT}}} \sum_{m \in [0, L_{\text{FFT}} - 1]^d} \sum_{j=1}^{N} f_j g_\tau(Lm/L_{\text{FFT}} - x_j) e^{-i2\pi k \cdot m / L_{\text{FFT}}} \tag{24}$$

This operation can be done in $\mathcal{O}(N_{\text{FFT}} \log(N_{\text{FFT}}))$ steps, independently of the number of inputs. Once this is computed, one can compute the Fourier transform of $f$ at each frequency point by

$$F(k) = \left(\frac{\pi}{\tau}\right)^{d/2} e^{\|k\|^2 \tau} F_\tau(k) \tag{25}$$

Once the Fourier transform of the Dirac delta train is ready, we multiply it by the Fourier multiplier $\hat{\phi}(k)$, which is the Fourier transform of $\phi$:

$$\hat{v}(k) = \hat{\phi}(k) F(k) \tag{26}$$

In the next sage, one needs to compute the inverse transform, and evaluate into the target points $\{x_i\}$. First we deconvolve the signal

$$\hat{v}_{-\tau}(k) = \left(\frac{\pi}{\tau}\right)^{d/2} e^{\|k\|^2 \tau} \hat{v}(k) \tag{27}$$

and compute the inverse Fourier transform

$$u_{-\tau}(x) = \sum_{k \in [0, N_{\text{FFT}}-1]^d} \hat{v}_{-\tau}(k) e^{ik \cdot x}. \tag{28}$$

Next, we interpolate to the point cloud

$$u(x_j) = u_{-\tau} * g_\tau(x_j) = \frac{1}{L^d} \int_{[0,L]^d} u_{-\tau}(x) g_\tau(x_j - x) \, dx \tag{29}$$

$$\approx \frac{1}{N_{\text{FFT}}} \sum_{m \in [0, L_{\text{FFT}}-1]^d} u_{-\tau}(Lm/L_{\text{FFT}}) \, g_\tau(x_j - Lm/L_{\text{FFT}}) \tag{30}$$

Even though in the current implementation all the parameters of the NUFFT are fixed, they can in principle be trained along with the rest of the networks. This training, if done naively increases significantly the computational cost. How to perform this operation efficiently is a direction of future research.

**Derivatives** For the computation of the forces in equation 5 one needs to compute the derivatives of the total energy $U^{\text{NN}}$ with respect to the inputs, in nearly-linear time. The main obstacle is how to compute the derivatives of the LRC-layer with respect to the point-cloud efficiently. To simplify the notation, we only discuss the case that $d = 1$, but the argument can be seamlessly extended to the case when $d > 1$.

Recall that $u_i = \sum_{j=1}^{N} \phi_\theta(x_i - x_j) f_j$, then the Jacobian of the vector $u$ with respect to the inputs is given by

$$(\nabla u)_{i,j} := \frac{\partial u_i}{\partial x_j} = \begin{cases} -f_j \phi_\theta'(x_i - x_j), & \text{if } j \neq i, \\ \sum_{k \neq i} f_k \phi_\theta'(x_i - x_k), & \text{if } j = i. \end{cases} \tag{31}$$

As it will be explained in the sequel, for the computation of the forces in equation 5 one needs to compute the application of the Jacobian of $u$ to a vector. For a fixed vector $v \in \mathbb{R}^N$, the product $(\nabla u) \cdot v$ can be written component-wise as

$$((\nabla u) \cdot v)_i = -\sum_{j \neq i} v_j f_j \phi_\theta'(x_i - x_j) + v_i \sum_{j \neq i} f_j \phi_\theta'(x_i - x_j),$$

$$= -\sum_{j=1}^{N} v_j f_j \phi_\theta'(x_i - x_j) + v_i \sum_{j=1}^{N} f_j \phi_\theta'(x_i - x_j),$$

where we have added $\pm v_i f_i \phi'(0)$ in the last equation and then distributed it within both sums. Let us define the following two long-range convolutions

$$w_i = -\sum_{j=1}^{N} v_j f_j \phi_\theta'(x_i - x_j), \qquad \text{and} \qquad p_i = \sum_{j=1}^{N} f_j \phi_\theta'(x_i - x_j), \tag{32}$$

each of which can be performed in $\mathcal{O}(N + N_{\text{FFT}} \log N_{\text{FFT}})$ steps using the NUFFT algorithm combined with the convolution theorem. In this case the derivative of $\phi$ can be computed numerically in the Fourier domain to a very high accuracy. Now one can leverage the expression above to rewrite $(\nabla u) \cdot v$ as

$$((\nabla u) \cdot v)_i = w_i + v_i p_i, \tag{33}$$

which can then be computed in nearly-linear time. The same is also true for $v \cdot (\nabla u)$.

## C.3 LONG-RANGE DESCRIPTOR

As mentioned before, the output of the LRC-layer is given by $\{u(x_i)\}_{i=1}^{N}$. For each particle we feed the output $u(x_i)$ to the long-range descriptor network $h_\theta : R \to R^{m_{1r}}$, whose structure is the same

as the local descriptor $f_\theta$ mentioned in appendix C.1 except that the activation function is taken to be ReLU. The long-range descriptor, defined as

$$\mathcal{D}^i_{\text{lr}} = g_\theta(u(x_i)) \tag{34}$$

for the particle $x_i$ is concatenated with the corresponding short-range descriptor (which it is itself the concatenation of two short-range descriptors) and fed together to the total fitting network $\mathcal{F} : \mathbb{R}^{m_{\text{sr}}+m_{\text{lr}}} \to \mathbb{R}$. The results are then added together to obtain the total energy

$$U^{\text{NN}} = \sum_{i=1}^{N} \mathcal{F}(\mathcal{D}^i_{\text{sr}}, \mathcal{D}^i_{\text{lr}}). \tag{35}$$

It is clear that the energy can be evaluated in nearly-linear complexity.

In what follows we show that the force computation is also of nearly-linear. For simplicity we focus on the one-dimensional network and assume that $K_{\text{chnls}} = 1$, $\mathcal{O}(m_{\text{sr}}) = \mathcal{O}(m_{\text{lr}}) = \mathcal{O}(1)$ and that the depth of the neural networks is $\mathcal{O}(1)$. As defined in the prequel the forces are given by $F^{\text{NN}} = -\nabla_x U^{\text{NN}}$, Which can be written component wise as

$$F^{\text{NN}}_j = -\partial_{x_j} U^{\text{NN}} = -\sum_{i=1}^{N} \left[ \partial_1 \mathcal{F}(\mathcal{D}^i_{\text{sr}}, \mathcal{D}^i_{\text{lr}}) \partial_{x_j} \mathcal{D}^i_{\text{sr}} + \partial_2 \mathcal{F}(\mathcal{D}^i_{\text{sr}}, \mathcal{D}^i_{\text{lr}}) g'_\theta(u_i) \partial_{x_j} u_i \right], \tag{36}$$

or in a more compact fashion as

$$F^{\text{NN}} = -\nabla U^{\text{NN}} = -\left( v_{\text{sr}} \cdot D_{\text{sr}} + v_{\text{lr}} \cdot \nabla u \right). \tag{37}$$

Here $v_{\text{sr}}$, and $v_{\text{lr}}$ are vectors defined component-wise as $(v_{\text{sr}})_i = \partial_1 \mathcal{F}(\mathcal{D}^i_{\text{sr}}, \mathcal{D}^i_{\text{lr}})$, and $(v_{\text{lr}})_i = \partial_2 \mathcal{F}(\mathcal{D}^i_{\text{sr}}, \mathcal{D}^i_{\text{lr}}) g'_\theta(u_i)$. In addition $(D_{\text{sr}})_{i,j} = \partial_{x_j} \mathcal{D}^i_{\text{sr}}$ and $\nabla u$ is defined above.

The first term in the right-hand side is easy to compute, given that $D_{\text{sr}}$ is sparse: the $i, j$ entry is non-zero only if the particle $x_i$ is in the interaction list of $x_j$. Given that the cardinality of the interaction list is bounded, $D_{\text{sr}}$ has $\mathcal{O}(N)$ non-zero entries in which each entry requires $\mathcal{O}(1)$ work, thus the first term in the right-hand side of equation 37 can be computed in $\mathcal{O}(N)$. At first glance the complexity of second term seems to be much higher. However, as discussed above, by using equation 33, we can apply the matrix (or its transpose) to a vector in $\mathcal{O}(N + N_{\text{FFT}} \log N_{\text{FFT}})$ time and the computation of vector $v_{\text{lr}}$ requires $\mathcal{O}(1)$ work per entry, thus resulting in a complexity of $\mathcal{O}(N + N_{\text{FFT}} \log N_{\text{FFT}})$ for computing the second term in equation 37. Finally, adding both contributions together results in an overall $\mathcal{O}(N + N_{\text{FFT}} \log N_{\text{FFT}})$ complexity for the forces.

To summarize, both the computation of the energy and the forces can be performed in $\mathcal{O}(N)$ time.

## C.4 TRAINING

We use the Adam optimizer (Kingma & Ba, 2015) along with an exponential scheduler. The learning rate with the initial learning rate taken to be 0.001 and, for every 10 epochs, it decreases by a factor of 0.95. In order to balance the computational time and the accuracy, a multi-stage training is adopted, where at each stage we modify the batch-size and the number of epochs. In particular, four stages are used: we start using a batch size of 8 snapshots and train the network 200 epochs and then at each stage we double both the size of the batch size and the number of epochs. In the two-scale training strategy, the same training parameters defined above are used for each stage.

## C.5 DEPENDENCY ON $N_{\text{FFT}}$

We measure the impact of $N_{\text{FFT}}$ on the approximation error, using a couple of examples in the one- and two-dimensional settings.

For the one-dimensional case, we test a screened-Coulomb type potential with parameters $\mu_1 = 5.0$, $\mu_2 = 0.5$, $\alpha_1 = 0.5$, $\alpha_2 = 0.5$, and $N_{\text{sample}} = 1000$. The domain $\Omega$ is $[0, 50]$ and $N = 200$. We run the one-scale training procedure with varying $N_{\text{FFT}}$ (the number of Fourier multipliers), starting from $N_{\text{FFT}} = 63$ and doubling them until $N_{\text{FFT}} = 501$. Table 5 shows that the errors are relatively insensitive to the value of $N_{\text{FFT}}$. The accuracy achieved by the architecture without the LRC-layer (denoted as None in Tables 5) is added in order to demonstrate that the architecture is indeed capturing the LRIs.

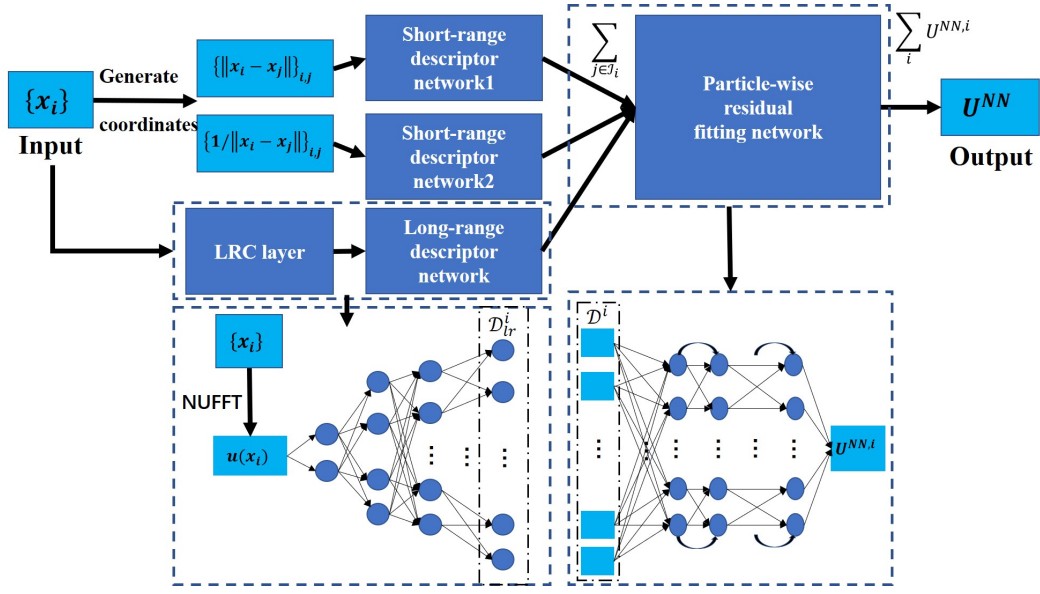

Figure 7: The structure of full-range network.

Table 5: Error with respect to $N_{\text{FFT}}$ in the 1D case

| $N_{\text{FFT}}$ | None | 63 | 125 | 251 | 501 |
|---|---|---|---|---|---|
| Relative testing error | 0.06143 | 0.00536 | 0.00546 | 0.00545 | 0.00539 |

For the two-dimensional case, a screened-Coulomb type potential is tested with $\mu_1 = 10.0$, $\mu_2 = 1.0$, $\alpha_1 = 0.9$, $\alpha_2 = 0.1$. Here $\Omega = [0,5]^2$, $N = 50$ and $N_{\text{sample}} = 1000$. Starting with $N_{\text{FFT}} = 21^2$, we steadily increase its value and repeat the same training procedure. The results are summarized in Table 6 where one observes the same trend as in the one-dimensional case.

Table 6: Error with respect to $N_{\text{FFT}}$ in the 2D case

| $N_{\text{FFT}}$ | None | $21^2$ | $31^2$ | $45^2$ | $63^2$ |
|---|---|---|---|---|---|
| Relative testing error | 0.01872 | 0.00202 | 0.00168 | 0.00153 | 0.00177 |

In addition, we recall that the Fourier multipliers are parametrized following

$$\hat{\phi}_{\beta,\lambda}(k) = \frac{4\pi\beta}{\|k\|^2 + \lambda^2}, \tag{38}$$

where $\beta$ and $\lambda$ are two trainable parameters with $\lambda$ providing a measure of the decay in space. Therefore, $N_{\text{FFT}}$ only determines the number of Fourier modes and not the parameters of the ansatz. As long as the Fourier kernel is properly sampled, the method is able to compute the correct characteristic interaction length.

One can observe this phenomenon in the experiment above, in which we extract the terminal value after training of the parameters $\lambda_1$ and $\lambda_2$ that correspond to the two channels in the LRC-layer, as summarized in Table 7. We observe that the value of $\lambda_2$ is very close to that of $\mu_2$, which is responsible for the LRIs even for small values of $N_{\text{FFT}}$.

Table 7: Values of parameters $\lambda_1$ and $\lambda_2$ after training with respect to $N_{\mathrm{FFT}}$.

| $N_{\mathrm{FFT}}$ | 1D case | | | | 2D case | | | |
|---|---|---|---|---|---|---|---|---|
| | 63 | 125 | 251 | 501 | $21^2$ | $31^2$ | $45^2$ | $63^2$ |
| $\lambda_1$ | 2.697 | 3.689 | 4.181 | 4.599 | 3.608 | 3.664 | 3.096 | 2.955 |
| $\lambda_2$ | 0.522 | 0.525 | 0.517 | 0.519 | 0.926 | 1.039 | 1.088 | 1.082 |

