# OpenReview forum: "Efficient Long-Range Convolutions for Point Clouds"
_ICLR.cc/2021/Conference — Reject_

### Official Review · AnonReviewer2 · 2020-10-13
**good improvement of existing method is presented, experiments could be improved**

**Rating:** 6
**Confidence:** 4

**Review:**

Edit: I was unaware that papers could be submitted to arXiv simultaneously, I am sorry for that. Here is my (very late) review.
I made it before reading other reviewers' reviews.

An efficient method for fitting long-range style interactions in point clouds is presented.
It makes use of NUFFT rather than convoluting a long-ranged (thus system-size and expensive) kernel directly in real space.
Along with this architecture, a method for training it efficiently is presented. This 2 steps strategy consists in training the short-range part of the kernels well with a lot of (supposedly inexpensive) short range data, and fitting the long-range kernels with less data in a second time.

Overall, the paper is clearly written and clearly exposes the methods used.
The results are interesting for applications but do not seem ground-breaking (although I am not an expert of point clouds -specialized networks).
In terms of experimental results, I think a couple of points would deserved to be answered (see below).

In conclusion, I think the paper is marginally above acceptance level.


In terms of results, the paper clearly shows that NUFFT scales essentially like O(N) (with N the points number) whereas naive direct space convolution scales as O(N^2).
However, when I read the algorithm (page 3), my main curiosity is : how well does the algorithm deals with large systems (large domain \Omega), and the competing parameters seems to be the resolution of the function g_\tau (which plays the role of mollified dirac) versus the system size \Omega or L. Concretely, I expect that for too large \tau/L (presented in appendix, Equation 20), the precision of the long range kernels will be poor and error will be large; and in the opposite case of very small \tau/L, precision will be good but compute time will increase (I guess it would increase as FFT does, in O(L/\tau log(L/\tau))).
Probably there is a regime where direct convolution (which does not need the approximation introduced by g_\tau, as far as I understood), is better then the NUFFT approach introduced here. I would guess that in the large system size (large \Omega=L^d) and small particles number N limit (i.e. the low density limit), the direct approach does well ?
I think such a discussion (and possibly a couple of experiments) would improve the paper a lot, showing the limits of the method and letting the reader understand the reasons for its strengths (which are very real, I do believe it !)


A bit more of commenting on the experiments' results would be appreciated. For instance, it seems that the 2-scale training strategy is especially efficient (not needing many samples) when the LRIs are sufficiently strong (figure 3, right). This is probably an effect of the "screening" of LRI by short-range Interactions when alpha_1 is tooo strong compared to alpha_2, making it harder to learn about LRIs.
Also, the fact that all curves essentially collapse beyond a given N_sample could receive a comment.
A couple of remarks like that about the strengths and limits of the approach would be nice.


Aside from these 2 main comments, I have minor remarks of presentation:
- figure 2 left and figure 3 both sides: poor choice of colors/linestyles. Curves come in pairs, and this should be suggested in the choice of display, using similar colors for pairs, and different linetyles in different pairs. It would improve readability (also, think of the color blind people!)
- fig2 , right: the O(N), O(N^2) scalings are un-readable. make them parallel to the plots you do and simple black dashed to improve clarity.

In the tables, although here results are "trivial" (always the same line has the smallest error), you could use the convention of putting in bold the better numbers.

Table 1, you say "mu_2 can be arbitrary here". What you mean is that it needs not be defined, because alpha_2=0 ?
I found this sentence confusing (maybe it's just me).

---

> ### Public Comment · ~Gautam_Kamath1 · 2020-11-13
> **Unfamiliar with the conference rules?**
>
> First, the CfP (https://iclr.cc/Conferences/2021/CallForPapers) for the conference explicitly allows posting to arxiv: "Submission of the paper to archival repositories such as arXiv is allowed."
>
> Second, your comment shines a light on the author identities, tempting other reviewers to click the link, thus ruining the spirit of ICLR's lightweight double blind process. This type of comment would have been better posted as a private comment to the AC to ask if should still be reviewed. A reviewer for one of my papers did something along these lines.
>
> Perhaps the Reviewer Guide (https://iclr.cc/Conferences/2021/ReviewerGuide) should have made note of these points.

---

> ### Author Response · Authors · 2020-11-24
> **Answers to comments (part1)**
>
> Thank you for your review and feedback, please find below the answers to your comments.
>
> *Edit: I was unaware that papers could be submitted to arXiv simultaneously, I am sorry for that. Here is my (very late) review. I made it before reading other reviewers' reviews.*
>
> *An efficient method for fitting long-range style interactions in point clouds is presented. It makes use of NUFFT rather than convoluting a long-ranged (thus system-size and expensive) kernel directly in real space. Along with this architecture, a method for training it efficiently is presented. This 2 steps strategy consists in training the short-range part of the kernels well with a lot of (supposedly inexpensive) short range data, and fitting the long-range kernels with less data in a second time.*
>
> *Overall, the paper is clearly written and clearly exposes the methods used. The results are interesting for applications but do not seem ground-breaking (although I am not an expert of point clouds -specialized networks). In terms of experimental results, I think a couple of points would deserved to be answered (see below).*
>
> *In conclusion, I think the paper is marginally above acceptance level.*
>
> *In terms of results, the paper clearly shows that NUFFT scales essentially like O(N) (with N the points number) whereas naive direct space convolution scales as O(N^2). However, when I read the algorithm (page 3), my main curiosity is : how well does the algorithm deals with large systems (large domain \Omega), and the competing parameters seems to be the resolution of the function g_\tau (which plays the role of mollified dirac) versus the system size \Omega or L. Concretely, I expect that for too large \tau/L (presented in appendix, Equation 20), the precision of the long range kernels will be poor and error will be large; and in the opposite case of very small \tau/L, precision will be good but compute time will increase (I guess it would increase as FFT does, in O(L/\tau log(L/\tau))). Probably there is a regime where direct convolution (which does not need the approximation introduced by g_\tau, as far as I understood), is better then the NUFFT approach introduced here. I would guess that in the large system size (large \Omega=L^d) and small particles number N limit (i.e. the low density limit), the direct approach does well ? I think such a discussion (and possibly a couple of experiments) would improve the paper a lot, showing the limits of the method and letting the reader understand the reasons for its strengths (which are very real, I do believe it !)*
>
> **Answer**\
> Thank you for your comment; this is an excellent point.
>
> We agree that if $\tau/L$ is very large, the underlying grid $N_{FFT}$ will be too small, and the error would increase due to an inadequate sampling of the kernel in the Fourier domain. Similarly, if $tau/L$ is too small, i.e., when $N_{FFT}$ is large, the computational cost would increase accordingly.
>
> We have added a new section in the Appendix, namely Appendix C.5, to illustrate this issue. The results in Appendix C.5 show that the current methodology is relatively insensitive to the density of the Fourier grid (and the underlying gridding in space). This means that in practice, we may be able to choose a relatively larger $\tau$, and hence a relatively small $N_{FFT}$ without sacrificing accuracy.
>
> We agree that the crossover between the NUFFT approach and the direct approach will be system dependent, and the crossover point is very worth understanding in the neural network context. However, since we received this comment at a relatively late stage of the review, we were not able to finish running the results for an in-depth study of this matter. Hence we plan to explore this direction more thoroughly in future work.

---

> > ### Author Response · Authors · 2020-11-24
> > **Answers to comments (part 2)**
> >
> > -------
> >
> > *A bit more of commenting on the experiments' results would be appreciated. For instance, it seems that the 2-scale training strategy is especially efficient (not needing many samples) when the LRIs are sufficiently strong (figure 3, right). This is probably an effect of the "screening" of LRI by short-range Interactions when alpha_1 is too strong compared to alpha_2, making it harder to learn about LRIs. Also, the fact that all curves essentially collapse beyond a given N_sample could receive a comment. A couple of remarks like that about the strengths and limits of the approach would be nice.*
> >
> >
> > **Answer**\
> > Thank you for the observation. Indeed, figure 3 (now figure 4 in the revised manuscript) seems to indicate that the two-scale strategy is more efficient when the LRI is stronger, in the sense that the test error is less sensitive with respect to the number of training samples. However, note that when the LRI is stronger, the test error is also slightly larger. Thus we cannot conclude that our training strategy becomes more efficient in the presence of strong LRIs. The qualitative behavior is also present in the 1D example (now figure 3 (left) in the revised manuscript). We have added some comments on page 9.
> >
> > The fact that the curve collapse is due basically to the training dynamics' stability, i.e., using different random seeds to train the network provides very similar accuracy. We have added several comments on pages 7 and 8 and some comments on Appendix C.5.
> >
> > --------
> >
> >
> > *Aside from these 2 main comments, I have minor remarks of presentation:*
> >
> > *figure 2 left and figure 3 both sides: poor choice of colors/linestyles. Curves come in pairs, and this should be suggested in the choice of display, using similar colors for pairs, and different linetyles in different pairs. It would improve readability (also, think of the color blind people!)*
> >
> >
> > **Answer**\
> > Thank you for your comment. We have modified the plots to improve readability.
> >
> > -------
> >
> >
> > *fig2 , right: the O(N), O(N^2) scalings are un-readable. make them parallel to the plots you do and simple black dashed to improve clarity.*
> >
> >
> > **Answer**\
> > Thank you for your comment. We have modified the plots.
> >
> > --------
> >
> > *In the tables, although here results are "trivial" (always the same line has the smallest error), you could use the convention of putting in bold the better numbers.*
> >
> > **Answer**\
> > Thank you for your comment we have, modified the Table and used a bold font for the best results.
> >
> > -------
> >
> > *Table 1, you say "mu_2 can be arbitrary here". What you mean is that it needs not be defined, because alpha_2=0 ? I found this sentence confusing (maybe it's just me).*
> >
> > **Answer**\
> > Thank you for your comment. What we meant, is that given that $\alpha_2$ is zero, $\mu_2$ can take any particular value. We have modified the phrasing of the sentence to avoid unnecessary confusion.

---

### Official Review · AnonReviewer1 · 2020-10-26
**An interesting work**

**Rating:** 6
**Confidence:** 3

**Review:**

The paper proposes an efficient long-range convolution method for point clouds by using the non-uniform Fourier transform. The long-range convolutional (LRC)-layer mollifies the point cloud to an adequately sized regular grid, computes its Fourier transform, multiplies the results by a set of trainable Fourier multipliers, computes the inverse Fourier transform, and finally interpolates the result back to the point cloud. The method is demonstrated to be effective by a N-body problem.

Overall, the paper is clearly written with high quality. The originality of the paper seems to be not very strong since it directly adapts the NUFFT to this work. Besides that, there are several concerns about this paper that need to be addressed:

1. How to choose the grid size L_{FFT} in the Fourier Space? How does this parameter affect the results?

2. The global pooling layers in DNN can also capture the long-range information to some extent, and are also very efficient. How does the LRC compare with the global pooling layers?

---

> ### Author Response · Authors · 2020-11-24
> **Answer to comments**
>
> Thank you for your review and feedback, please see below the answers to your comments.
>
> *The paper proposes an efficient long-range convolution method for point clouds by using the non-uniform Fourier transform. The long-range convolutional (LRC)-layer mollifies the point cloud to an adequately sized regular grid, computes its Fourier transform, multiplies the results by a set of trainable Fourier multipliers, computes the inverse Fourier transform, and finally interpolates the result back to the point cloud. The method is demonstrated to be effective by a N-body problem.*
>
> *Overall, the paper is clearly written with high quality. The originality of the paper seems to be not very strong since it directly adapts the NUFFT to this work. Besides that, there are several concerns about this paper that need to be addressed:*
>
>
> **Answer**\
> Thank you for your comments. The contribution of this paper is two-fold:
> - the introduction of the NUFFT to problems involving point-cloud, which to our knowledge has not appeared in the literature.
> - the introduction of a two-scale training procedure that could be helpful in the context of large scale simulation, where one can only have access to a limited number of snapshots.
>
> Even though we agree that the NUFFT is itself a well-known technique, it has not been used to accelerate the performance of neural network layers in machine learning problems. Our work introduces this method to a broader audience and provides a systematic way to treat long-range interactions, which is competitive with state-of-the-art localized networks.
>
> -------
>
>
> *How to choose the grid size L_{FFT} in the Fourier Space? How does this parameter affect the results?*
>
> **Answer**\
> Thank you for your comment. This is an excellent question. We have added a new section in the Appendix, namely Appendix C.5 to illustrate this issue. The results in Appendix C.5 show that the current methodology is relatively insensitive to the density of the Fourier grid (and the underlying gridding in space). We can expect that when $L_{FFT}$ is small, i.e., when we have a coarse gridding resolution, the error would increase due to the inability to sample the kernel Fourier domain adequately. However, due to the smooth parametrization of the kernel in Fourier domain, we can use a relatively small $L_{FFT}$ with a small penalty to the accuracy, even if we severely reduce $N_{FFT}$ as shown in the newly added Appendix C.5. In a nutshell, as long as the parametrized kernel is properly sampled in Fourier domain, the method can efficiently capture the LRIs.
>
> Besides Appendix C.5, we have added a couple of comments on page 4.
>
> -------
>
>
> *The global pooling layers in DNN can also capture the long-range information to some extent, and are also very efficient. How does the LRC compare with the global pooling layers?*
>
>
> **Answer**\
> Thank you for your comment. Yes, global pooling layers can capture long-range information, particularly when the output is a single number. In our case, however, we suppose that both that input and output are point-clouds, and, to our knowledge, there is no standard implementation of global pooling layers in this context. One non-standard fashion of implementing a global pooling layer in this context is the short-range neural network that we use as a comparison. In a nutshell, we can think of the short-range neural network in Eq. 5 as the composition of a very non-linear convolution layer followed by a global pooling layer, as shown in Eq. 3, which outputs the energy, plus a derivative of the output with respect to the inputs, which produces the forces. As shown in the experiments, in particular, Tables 1, 2 and 3, long-range interactions can not be seamlessly captured, even when using global pooling layers.
>
> To make this clearer, we added a sentence on page 3 to make sure that we are focusing on problems in which the input and output are point clouds; thus, traditional global pooling layers may not be readily applicable. Also, we added a small paragraph on page 6, which provides a concise interpretation of the short-range network as a non-linear graph-convolution followed by a global pooling layer.

---

### Official Review · AnonReviewer3 · 2020-10-28
**This paper needs to make changes in some aspects.**

**Rating:** 5
**Confidence:** 4

**Review:**

This paper concentrated on exploring how to efficiently extract the interaction information from the point clouds.  A key point lies in utilizing the non-uniform Fourier transform, rather than the regular Fourier transform. However, there exist some issues that need to be solved.

+ves:
+ The exploration of long-range interactions for point clouds is interesting.

+ The paper is well written. The related work makes a clear description of many fields about point cloud.

Concerns:
1. In the introduction part, the authors describe many tasks that rely on point-cloud presentation. However, the authors ignore pointing out the existing issues. The authors should clearly present them.

2. In the algorithmic section, the authors claim that "NUFFT serves as the corner-stone of the LRC-layer". So "NUFFT" is the unique solution? Whether or not some operations can replace FFT?

3. Point-cloud is indeed important for many tasks as described in the introduction part, but the authors just explore the effects of the proposed algorithm in a "synthetic" experiment. The experimental results are not convincing for readers, the authors should conduct more real-world tasks to verify the effectiveness of the proposed method.

4. The presentation of this work needs to improve, if possible, the authors should provide an intuitive schematic diagram to present the procedure of proposing this idea.

5. In the experimental section, the author should replot Figure2-3 to ensure clear enough for a better read.

#########################################################################

Minor Comments:

(1)  “N_sample” and "N" maybe exist the inclusion relation，it will be better to replace one of them with the other form;

(2)  The formula system is a little vague, if possible, the authors can simplify them to clearly describe.

---

> ### Author Response · Authors · 2020-11-24
> **Answers to comments**
>
> Thank you for your review and feedback.
>
>
> *Concerns:*
>
> *1. In the introduction part, the authors describe many tasks that rely on point-cloud presentation. However, the authors ignore pointing out the existing issues. The authors should clearly present them.*
>
>
> **Answer**\
> Thank you for raising this question. In a nutshell, we are aiming to obtain a long range convolutional layer that satisfies four requirements:
> -it can seamlessly respect symmetries, such as rotation, translation and permutation invariance,
> -it has a nearly linear cost with respect the number of input/output dimension
> -it can generalize to bigger problems seamlessly
> -it can capture long-range interactions accurately.
>
> In particular, machine learning methods can satisfy two or perhaps three items on our wish list, but so far, to the best of our knowledge no method can satisfy all of them. We have added several comments in the introduction to discuss explicitly the issues above.
>
> ----------
>
> *2. In the algorithmic section, the authors claim that "NUFFT serves as the cornerstone of the LRC-layer". So "NUFFT" is the unique solution? Whether or not some operations can replace FFT?*
>
>
> **Answer**\
> Thank you for your comment. You are right; NUFFT is the cornerstone of the LRC-layer presented in this paper. One can use, e.g., the fast multipole method (FMM) or other fast summation methods to achieve nearly-linear time. However, we find that using the NUFFT makes the implementation more straightforward, given that we only need to train the Fourier multiplier while keeping the rest of the operations unchanged. On the other hand, when using fast summation methods such as FMM, one needs to change the full algorithm if the kernel $\phi_{\theta}$ changes. We added a paragraph and a footnote on page 3 to further explain this point.
>
> -----------
>
> *3. Point-cloud is indeed important for many tasks as described in the introduction part, but the authors just explore the effects of the proposed algorithm in a "synthetic" experiment. The experimental results are not convincing for readers, the authors should conduct more real-world tasks to verify the effectiveness of the proposed method.*
>
> **Answer**\
> Thank you for your comment. Our goal is indeed to develop an efficient strategy to model LRIs in real chemical and materials systems. This is the first work along this line, and we selected the model N-body potential as the target application in order to unambiguously demonstrate that 1) the computational scaling is nearly linear, and 2) the sample complexity for large scale data can be reduced. The latter is particularly important in molecular modeling, since the training data are usually obtained from first principle electronic structure calculations, and the large scale data are very expensive to obtain.  We expanded the discussion along this line both in the introduction (page 3) and the conclusion (page 9).
>
> ------
>
> *4. The presentation of this work needs to improve, if possible, the authors should provide an intuitive schematic diagram to present the procedure of proposing this idea.*
>
> **Answer**\
> Thank you for your comment. We have added a diagram of the algorithmic pipeline of the LRC-layer, in particular, in the revised manuscript we illustrate in Fig. 1, how the NUFFT computes the Fourier Transform, which is the cornerstone of our instance of the LRC-layer.
>
> ---------
>
> *5. In the experimental section, the author should replot Figure2-3 to ensure clear enough for a better read.*
>
> **Answer**\
> Thank you for your comment. We have redrawn those plots to increase readability.
>
> ---------
>
>
> #########################################################################
>
> *Minor Comments:*
>
> *(1) "N_sample" and "N" maybe exist the inclusion relation，it will be better to replace one of them with the other form;*
>
>
> **Answer**\
> Thank you for your comment. Actually they are different: $N_{sample}$ is the number of snapshots used for training, whereas, $N$ is the number of particles (or points in the cloud point) in each snapshot. We have added a comment in the last paragraph in page 7 making the distinction clear.
>
> ---------
>
>
> *(2) The formula system is a little vague, if possible, the authors can simplify them to clearly describe.*
>
>
> **Answer**\
> Thank you for your comment. We added a table listing all the symbols and their meaning in the appendix.

---

### Official Review · AnonReviewer4 · 2020-10-29
**Utilizing the nonuniform FFT, a long range convolutional layer (LRC) is presented. A neural network which combines both LRC and short range conv layers are built. In a two-scale training strategy, first many small-scale data are trained without the LRC, then a small set of large-scale data are trained with both short and long range conv layers. The model is tested on 1D and 2D screened Coulomb particle configurations to estimate the overall potential energy and forces.**

**Rating:** 5
**Confidence:** 4

**Review:**

The paper is clearly written, and presents an approach to efficiently utilize long range convolutions through a nonuniform FFT in for coulomb particle configurations.

Pros:
- Presents a long range convolution layer, with an efficient implementation so that a neural network model can benefit from both short and long range interactions among data points.

- To model pairwise relations between points of the N-body potential, two short range descriptor networks are utilized, where the input relations are constructed both repulsively and attractively.

Cons:
Experimentally, the new long-short range NN model is validated only for computing the energy and forces for a N-body model. There are no experiments with real-life point cloud data.

In the proposed algorithm, a sampling in a regular grid is required. A fixed size Cartesian grid is utilized in the experiments, therefore, the effect of the gridding resolution is not analyzed.

The quantitative experimental results in Tables 1, 2, and 3 dwell on varying the screening coefficient mu, hence investigates how the test error changes for point configurations with varying screening parameters, which signifies the amount of diffusion of the potential. It can be seen that the test error for the proposed approach falls as the screening gets higher, that is when we have more localized interactions between the particles. What could be the significance of this finding for a practical application is not discussed, therefore not made clear.

The authors state that the proposed method could be a useful tool in a wide range of Machine Learning tasks. However, in the paper, only estimation of potential energy for Coulomb particle configurations is demonstrated as an application. I am curious to see how the presented approach could find use in learning tasks of real point cloud configurations.

The paper presents a way to incorporate long range convolutions in neural networks, which could be beneficial. However, experimental evaluation is not satisfactory, as immediate implications in point cloud analysis is not obvious.

---

> ### Author Response · Authors · 2020-11-24
> **Answers to comments (part 1)**
>
> Thank you for your review and feedback, please find below the answer to your comments.
>
> *The paper is clearly written, and presents an approach to efficiently utilize long range convolutions through a nonuniform FFT in for coulomb particle configurations.*
>
> *Pros:*
>
> *Presents a long range convolution layer, with an efficient implementation so that a neural network model can benefit from both short and long range interactions among data points.*
> *To model pairwise relations between points of the N-body potential, two short range descriptor networks are utilized, where the input relations are constructed both repulsively and attractively.*
>
> *Cons: Experimentally, the new long-short range NN model is validated only for computing the energy and forces for a N-body model. There are no experiments with real-life point cloud data.*
>
> **Answer**\
> Thank you for your comment. Please find below our response to the comment above, as well as  the question
>
> "The paper presents a way to incorporate long range convolutions in neural networks, which could be beneficial. However, experimental evaluation is not satisfactory, as immediate implications in point cloud analysis is not obvious."
>
> Our goal is indeed to develop an efficient strategy to model LRIs in real chemical and materials systems. This is the first work along this line, and we selected the model N-body potential as the target application in order to unambiguously demonstrate that 1) the computational scaling is nearly linear, and 2) the sample complexity for large scale data can be reduced. The latter is particularly important in molecular modeling, since the training data are usually obtained from first principle electronic structure calculations, and the large scale data are very expensive to obtain.  We expanded the discussion along this line both in the introduction (page 3) and the conclusion  (page 9).
>
> Concerning the real-life point cloud data, the evaluation of N-body potential is a foundational component in molecular modeling. It takes the atomic positions as the input point cloud, and the output (local potential and force) can also be viewed as point clouds. We expect the LRC-layer to become a useful component in designing neural networks for modeling real chemical and materials systems, where the LRI cannot be accurately captured using short ranged models.
>
> -------
> *In the proposed algorithm, a sampling in a regular grid is required. A fixed size Cartesian grid is utilized in the experiments, therefore, the effect of the gridding resolution is not analyzed.*
>
> **Answer**\
> Thank you for your comment. This is an excellent question. We have added a new section in the Appendix, namely Appendix C.5 to illustrate this issue. The results in Appendix C.5 show that the current methodology is relatively insensitive to the density of the Fourier grid (and the underlying gridding in space). We can expect that when $L_{FFT}$ is small, i.e., when we have a coarse gridding resolution, the error would increase due to the inability to sample the kernel Fourier domain adequately. However, due to the smooth parametrization of the kernel in Fourier domain, we can use a relatively small $L_{FFT}$ with a small penalty to the accuracy, even if we severely reduce $N_{FFT}$ as shown in the newly added Appendix C.5. In a nutshell, as long as the parametrized kernel is properly sampled in Fourier domain, the method can efficiently capture the LRIs.
>
> Besides Appendix C.5, we have added a couple of comments on page 4.

---

> > ### Author Response · Authors · 2020-11-24
> > **Answers to comments (part 2)**
> >
> >
> > *The quantitative experimental results in Tables 1, 2, and 3 dwell on varying the screening coefficient mu, hence investigates how the test error changes for point configurations with varying screening parameters, which signifies the amount of diffusion of the potential. It can be seen that the test error for the proposed approach falls as the screening gets higher, that is when we have more localized interactions between the particles. What could be the significance of this finding for a practical application is not discussed, therefore not made clear.*
> >
> > **Answer**\
> > Thank you for your comment.
> >  The main message of Tables 1, 2 and 3 is to study the behavior of the error as the interaction length becomes larger. Even though this is a potential similar to a diffusion equation, we don't consider the tuning \mu to tune the diffusion. Instead we use it to tune the interaction length between different particles, i.e., in the computation of the energy and forces in Eq. 2, how many particles actually contribute to the calculation at each point. Thus, as we reduce mu, i.e.,  as we increase the characteristic interaction length, the number of particles contributing to the Energy and Forces increase.
> > Thus in Tables 1, 2, and 3 we observe that the accuracy of local networks decreases as this interaction length increases, however, we can be recover the accuracy by using the new architecture, which consists of a local network plus one single LRC-layer, which accounts for the LRIs.
> >
> > Perhaps the most direct consequence of this paper related to the molecular modeling community. So far applications of machine learning to molecular dynamics have mostly considered short localized interaction. This works has the potential to broaden the spectrum of application to more system in which the long-range interactions can not be easily neglected.
> >
> > We have added a paragraph in the introduction, as well as the conclusion to clarify these points.
> >
> > --------
> >
> > *The authors state that the proposed method could be a useful tool in a wide range of Machine Learning tasks. However, in the paper, only estimation of potential energy for Coulomb particle configurations is demonstrated as an application. I am curious to see how the presented approach could find use in learning tasks of real point cloud configurations.*
> >
> > **Answer**\
> > Thank you for the comment. This is the first work along this line, and we indeed expect that the LRC-layer can be useful in other contexts, such as machine learning tasks of regression and classification types. One of our current research direction is to employ the LRC layer for image classification processing tasks, and we will report the performance in a future work.

---

### Decision · Program_Chairs · 2021-01-10
**Final Decision**

**Decision:**

Reject

**Comment:**

This work proposes an efficient method for modelling long-range connections in point-cloud data. Reviewers found the paper to be generally well-written. On the less positive side, reviewers felt that the novelty of the work was marginal, and that the experimentation, limited to synthetic data in one domain, was too limited. These concerns remain after the discussion phase. In addition, the authors stated during the discussion that "Our goal is indeed to develop an efficient strategy to model LRIs in real chemical and materials systems”, which conflicts with the presentation of the work as motivated by more general point cloud modelling problems. Given these weaknesses, the final decision was to reject.